# Oxidative stress response in children undergoing cardiac surgery: Utility of the clearance of isoprostanes

Stephanie Hadley[1], Debora Cañizo Vazquez[2], Miriam Lopez Abad[3], Stefano Congiu[4], Dmytro Lushchencov[5], Marta Camprubí Camprubí[2]*, Joan Sanchez-de-Toledo[6]

1 Vanderbilt University School of Medicine, Nashville, TN, United States of America, 2 BCNatal, Barcelona Center for Maternal Fetal and Neonatal Medicine, Hospital Sant Joan de Déu and Hospital Clinic, University of Barcelona, Barcelona, Spain, 3 Laboratory Sciences, Fundació Sant Joan de Déu, Barcelona, Spain, 4 Department of Cardiothoracic Surgery, Hospital Sant Joan de Déu, Barcelona, Spain, 5 Department of Anesthesia, Hospital Sant Joan de Déu, Barcelona, Spain, 6 Department of Cardiology, Hospital Sant Joan de Déu, Barcelona, Spain

* mcamprubic@sjdhospitalbarcelona.org

## Abstract

### Introduction

Cardiac surgery (CS) in pediatric patients induces an overt oxidative stress (OS) response. Children are particularly vulnerable to OS related injury. The immaturity of their organs and antioxidant systems as well as the induction of OS in cardio-pulmonary bypass (CPB) surgery may have an important impact on outcomes. The purpose of this study was to describe the OS response, measured by urinary free 8-iso-PGF2α, in infants undergoing CS and to evaluate the relationship between OS response and post-operative clinical outcomes.

### Methods

Infants with congenital heart disease undergoing CS with or without CPB were eligible for enrollment. Children were classified as neonates (<30 days) or infants (30 days—6 months) based on the age at surgery. Perioperative continuous non-invasive neuromonitoring included amplitude-integrated electroencephalogram and cerebral regional oxygen saturation measured with near-infrared spectroscopy. Urine 8-iso-PGF2α levels were measured before, immediately post-, and 24-hours post-surgery, and the 8-iso-PGF2 clearance was calculated.

### Results

Sixty-two patients (60% neonates) were included. Urine 8-iso-PGF2α levels 24 hours after surgery (8.04 [6.4–10.3] ng/mg Cr) were higher than pre-operative levels (5.7 [4.65–7.58] ng/mg Cr) (p<0.001). Those patients with a severe degree of cyanosis caused by Transposition of the Great Arteries (TGA) had the highest post-operative 8-iso-PGF2α levels. Patients with intra-operative seizures had higher post-operative 8-iso-PGF2α levels. 8-iso-PGF2α clearance at 24 hours post-surgery was different between newborns and infant patients, and it was inversely correlated with days of mechanical ventilation (p = 0.05), ICU LOS (p = 0.05) and VIS score at 24 hours (p = 0.036).

**Data Availability Statement:** All relevant data are within the manuscript and its Supporting information files.

**Funding:** This work was supported by a private grant from Godia and by a grant from Instituto Carlos III [PFIS17-144] (JSDT, MCC). The funders had no role in study design, data collection and analysis, decision to publish, or preparation of the manuscript.

**Competing interests:** The authors have declared that no competing interests exist.

**Abbreviations:** OS, Oxidative stress; CPB, Cardiopulmonary bypass; 8-iso-PGF2α, 8-iso-Prostaglandin-F2α; LOS, Length of stay; ICU, Intensive care unit; CHD, Congenital heart disease; TGA, Transposition of the great arteries; cSO2, Cerebral regional oxygen saturation; NBO, Newborn behavioral observations; aEEG, Amplitude-integrated electroencephalogram.

## Conclusions

Children undergoing CS, particularly neonatal patients, experience a significant post-operative OS response that might play an important role in postoperative morbidity. TGA patients undergoing arterial switch operations demonstrate the highest post-operative OS response. Rapid clearance of isoprostanes, which occurs more frequently in older patients with more mature antioxidant systems, might be associated with better clinical outcomes.

## Introduction

Under normal conditions, the body exists in a balance between oxidant and anti-oxidant substances. This equilibrium can be disturbed in many situations that cause increased free radical production, a phenomenon known as the oxidative stress (OS) response.

The pathophysiology behind the OS response observed during pediatric cardiac surgery, particularly in cases involving cardiopulmonary bypass (CPB), is complex and multi-factorial. Bypass circuits contribute to systemic inflammation through red blood cell contact with the circuit surfaces, the generation of mechanical shear stress, and significant hemodilution [1]. Each of these processes trigger increased OS through the generation of cytokines, complement system activation, and depletion of antioxidant defenses. Moreover, perioperative conditions of infants with congenital heart disease (CHD) such as hemodynamic instability, low cardiac output and cyanosis could induce endothelial cell damage and cause ischemia-reperfusion injury, conditions that have also been associated with increased OS [2]. In adults, each of these factors has been shown to contribute to several adverse outcomes in the postoperative period [3], but there is limited evidence in children. Children may be more vulnerable to OS injury, not only attributable to cardiopulmonary bypass [4], but also because of the immaturity of many of their organs and antioxidant systems, especially in newborns [5]. Though OS has been shown to increase following cardiac surgery in pediatric patients, its relationship to clinical outcomes remains uncertain [6, 7].

8-isoprostaglandin-F2α (8-iso-PGF2α) is the product of nonenzymatic, free radical-catalyzed peroxidation of arachidonic acids. Given its accuracy as an indicator of lipid peroxidation and its clinical stability, it is considered one of the most reliable biomarkers of OS [8].

The purpose of this study was *i)* to describe the variation of OS levels, measured by urinary free 8-iso-PGF2α, in infants undergoing cardiac surgery and *ii)* to evaluate the relationship between OS levels during cardiac surgery and post-operative clinical outcomes.

## Material and methods

### Patient management and clinical data

The study was reviewed and approved by the Hospital Sant Joan de Déu (Esplugues de Llobregat, Spain) Ethics Committee (PIC 120–17) and was conducted in accordance with the Declaration of Helsinki. Written informed consent was obtained for all subjects. This was a prospective, longitudinal study conducted at a tertiary referral university hospital.

All children with CHD up to 6 months of age undergoing cardiac surgery with or without CPB were eligible for enrollment. Exclusion criteria included proven or clinically suspected genetic syndrome, history of birth asphyxia or pre-existing brain damage.

Patients were classified depending on age at surgery, divided into neonatal (< 30 days) and infant (30 days—6 months) groups. Patients were further sub-classified as having undergone cardiac surgery with or without CPB. Cyanotic congenital heart diseases (CCHD) were also divided into *i)* CCHD with decreased blood flow ("typical" cyanosis, i.e. Fallot physiology) and

*ii)* CCHD with increased blood flow ("extreme" cyanosis from Transposition of Great Arteries (TGA) physiology) [9]. Balloon atrioseptostomy (BAS) was indicated when the interauricular communication had a restrictive flow.

Clinical data including prostaglandin use, preoperative management with sub atmospheric gas therapy, BAS requirement, days of mechanical ventilation, intensive care unit (ICU) length of stay (LOS), and overall LOS were recorded. Vasoactive-inotropic scores (VIS) within 24 hours post-surgery were calculated according to the Gaies et al. criteria [10].

**Operative management.**   Anesthetic management followed institutional cardiac anesthesia protocols with high-dose fentanyl, inhaled isoflurane and muscle relaxants. No benzodiazepines or barbiturates were administered during surgery. For infants undergoing CPB, an estimated flow of 2.8–3.0 L/m$^2$ BSA (175–200 mL/kg) after bicaval cannulation was maintained in order to keep mean radial and/or femoral arterial pressures in the range of 35–45 mmHg. Target ambient temperature during CPB was 22–34˚C. Deep hypothermia (16-18˚C) was used in cases with circulatory arrest. Selected anterograde cerebral perfusion was not routinely performed in our center in patients undergoing circulatory arrest. Continuous standard hemofiltration was performed during rewarming and weaning from bypass to achieve a target hematocrit between 28 and 31. No modified ultrafiltration at the end of the procedure was performed at our center. All patients received milrinone as standard early post-operative therapy combined with dopamine and epinephrine if needed. Postoperative analgesia and sedation were achieved with continuous infusions of morphine and dexmedetomidine.

**Newborn behavioral observations.**   Patients under 2 months of age were evaluated using the Newborn Behavioral Observations (NBO) tool to test neurobehavior prior to hospital discharge. The NBO was applied by a certified psychologist when medically feasible, at least five days after surgery and when the patients were not receiving sedative medications. Clinical interpretation was performed globally as well as in several sub-domains (Habituation, Attention, Arousal, Regulation, Handling, Quality of Movement, and Stress) [11].

## Perioperative neuromonitoring

A continuous amplitude-integrated electroencephalogram (aEEG) (NicoletOneTM, Natus, Middleton, WI) was used to monitor brain electrical activity during surgery and the post-operative period. Two neonatologists trained in aEEG interpretation blindly reviewed tracings.

Background patterns were analyzed in accordance to Hellström-Westas [12] classifications. They were classified as continuous normal voltage (CNV), discontinuous normal voltage (DNV), burst suppression (BS) and continuous low voltage. CNV and DNV were both considered normal, as even healthy infants may show discontinuous activity during sleep. Electrographic seizures were also identified.

For patients experiencing intra-operative seizures, the total seizure burden was calculated in minutes.

Cerebral regional oxygen saturation (cSO2) was measured with near-infrared spectroscopy (NIRS) using an INVOS 5100C Cerebral/Somatic Oximeter (Medtronic, Minneapolis, MN). Two appropriately sized transcutaneous monitors were placed on the patient's forehead before surgery and remained for up to 72 hours post-surgery. Data were exported in 30-second interval averages. Minutes spent <40% cSO2, <50% cSO2, outside of the 50–70% cSO2 range, and >85% cSO2 were calculated.

## Urine oxidative stress biomarkers

Urine samples were collected immediately before surgery, upon arrival to the ICU, and at 24 hours post-surgery. Samples were stored at -80˚C until analysis. Free urine concentrations of

8-iso-PGF2α, the product of nonenzymatic, free radical-catalyzed peroxidation of arachidonic acids, were quantified using an enzyme-linked immunoassay (Cell BioLabs, Inc., San Diego, CA). Levels were adjusted for urinary creatinine excretion and expressed in ng/mg of creatinine (ng/mg Cr).

8-iso-PGF2α clearance was defined as the percent change in urine 8-iso-PGF2α after 24 hours from surgery. It was calculated by using the following formula:

[(Post-surgery 8-iso-PGF2α– 24 hours post-surgery 8-iso-PGF2α) *100]/ Post-surgery 8-iso-PGF2α

## Statistical analysis

Study data were stored using the institution's Redcap electronic data capture tools. Statistical analyses were performed using SPSS version 25 (IBM, Armonk, New York) and STATA v13 package.

Variables were evaluated for normality and homogeneity of variance. Continuous variables are expressed as mean and standard deviation for normally distributed values and median and interquartile range for non-normally distributed values. Continuous variables were compared between two independent samples using the Student's t-test or the Mann Whitney U test based on distribution. Continuous variables were compared between multiple independent samples using the Kruskal-Wallace test. Correlations were performed using the Spearman correlation coefficient. Univariate and multivariate analyses were used for identifying relations between variables. Confounding factors were analyzed using linear regression. Statistical significance was considered when $p < 0.05$.

## Results

### Patients characteristics

Sixty-two patients undergoing cardiac surgery between November 2017 and February 2019 were included. Median age at the time of surgery was 21 days [IQR 8–96]. Thirty-seven (60%) patients were neonates. Patient's characteristics are displayed in Table 1.

All Infant patients underwent CPB surgery (n = 25). Neonatal patients were divided into sub-groups based on the type of surgery: CPB (n = 12) and non-CPB (n = 25). Twenty-five (40%) patients had no cyanosis, 26 (42%) had typical cyanosis, and 11 (18%) had extreme cyanosis caused by TGA. To further support these cyanosis classifications, pre-surgical and post-surgical percentages of hemoglobin saturation and partial pressures of oxygen were analyzed. Patients with no cyanosis had a mean percent hemoglobin saturation of 98.6% (+/-1.4), those with mild cyanosis had a mean of 91.5% (+/-7.38) and patients with TGA 79.6% (+/-7.9) (p = 0.001). Pre-surgical paO2 levels were also analyzed. Patients with no cyanosis had a mean of 145 mmHg (+/-65), patients with mild cyanosis had a mean of 82 mmHg (+/- 49) and patients with TGA 48.3 mmHg (+/-9.2) (p = 0.001), as demonstrated in Fig 1A. Post-surgical percentages of hemoglobin saturation and partial pressures of oxygen were not statistically different amongst groups (p = 0.1041 and 0.2399, respectfully), as seen in Fig 1B.

Only 4 patients required surgery with deep hypothermic circulatory arrest. Individual diagnoses are displayed in Table 2.

### Perioperative neuromonitoring results

Twenty-four (35.5%) patients demonstrated abnormal intra-operative electroencephalographic background patterns, and 13 (20%) experienced electrographic seizures during surgery.

**Table 1. Demographics and clinical characteristics of the study population.**

| Patient Characteristics | Infant Patients (n = 25) | Neonatal Patients (n = 37) | p-value |
|---|---|---|---|
| Male sex | 13 (52%) | 23 (62.1%) | 0.42 |
| Prematurity (<37 weeks) | 5 (6.45%) | 4 (8.06%) | 0.314 |
| Gestational age (weeks) | 38.9 [38.5–39.3] | 38.1 [37.36–38.9] | 0.972 |
| Birth weight (kg) | 3056 [2912–3200] | 2754 [2523–2984] | 0.0209 |
| Presurgical Prostaglandin infusion | 10 (27.78%) | 53 (80.3%) | 0.0001 |
| Sub-atmospheric therapy | 2 (5.5%) | 14 (22.22%) | 0.030 |
| BAS | 0 (0%) | 5 (7.9%) | 0.0001 |
| Age at surgery [days] | 106 [63–161] | 9 [7–14] | 0.001 |
| Cardiopulmonary Bypass | 25 (100%) | 12 (32%) | 0.001 |
| Coarctectomy | - | 13 (35%) | <0.001 |
| Other | - | 12 (32%) | <0.001 |
| STAT category* | | | 0.34 |
| 1 | 11 (44%) | 12 (32%) | |
| 2 | 7 (28%) | 14 (38%) | |
| 3 | 5 (20%) | 3 (8%) | |
| 4 | 2 (8%) | 7 (19%) | |
| 5 | - | 1 (3%) | |
| Biventricular repair | 22 (88%) | 29 (78%) | 0.33 |
| Left ventricular outflow tract obstruction | 4 (16%) | 17 (46%) | 0.015 |
| Mechanical ventilation (days) | 0 [0–2] | 2 [1–3] | 0.76 |
| ICU length of stay (days) | 5 [3–8] | 14 [9–28] | 0.073 |
| Hospital length of stay (days) | 8 [6–13] | 21 [14–40] | 0.06 |

*STS-EACTS Congenital Heart Surgery Mortality Categories.

Abnormal intra-operative background patterns were more frequent among newborn patients (41.46% vs 8.54%) (p = 0.001). No differences were detected in incidence of electroencephalographic seizures between newborn and infant patients (p = 0.148). Patients that underwent CPB surgery had more electroencephalographic seizures (p = 0.001), but no differences in background patterns were detected between CPB patients and no CPB patients (p = 0.797).

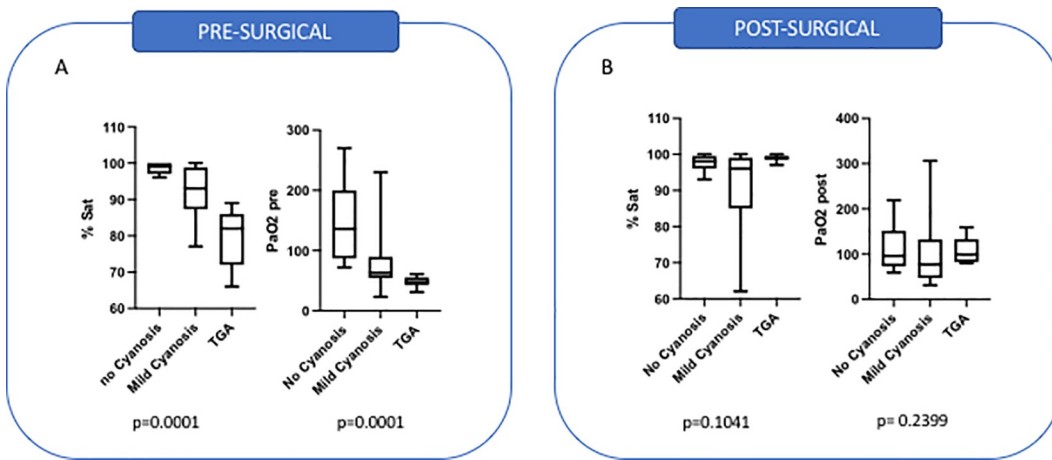

**Fig 1. A**: Pre-surgical levels of hemoglobin saturation and paO2. **B**: Post-surgical levels of hemoglobin saturation and paO2.

**Table 2. Patient diagnoses.**

| Type of defect | Type of surgery | Newborn/infant | N |
|---|---|---|---|
| Coarctation of the Aorta | 2 | 13/0 | 13 |
| Transposition of the Great Arteries | 1 | 11/0 | 11 |
| Ventricular Septal Defect | 1 | 0/10 | 10 |
| Tetralogy of Fallot | 1 | 2/8 | 10 |
| Pulmonary Atresia | 2/1 | 3/2 | 5 |
| Double Outlet Right Ventricle | 2/1 | 2/2 | 4 |
| Truncus Arteriosus | 1 | 2/1 | 3 |
| Hypoplastic Left Heart Syndrome | 1 | 2/0 | 2 |
| Total Anomalous Pulmonary Venous Return | 1 | 2/0 | 2 |
| Aortic Stenosis | 1 | 0/1 | 1 |
| Mitral Stenosis | 1 | 0/1 | 1 |

Type of surgery: CPB 1; no CPB 2.

The median cSO2 after surgery was 62% [59–65] and 63% [61–66] during the first 24 post-operative hours in CPB and post-CPB groups, respectively. Intraoperatively, patients who had surgery under CPB experienced higher cSO2 values (68% [63–73]) when compared to those with surgery without CPB (60% [57–62]) (p = 0.001). Patients undergoing circulatory arrest had the lowest cSO2 values. Differences in cSO2 values between CPB and no CPB patients are expressed in Table 3.

## Isoprostane measurements

The overall median level of 8-iso-PGF2α increased from 5.7 [4.65–7.58] ng/mg Cr before surgery to 10.5[8.7–12.4] ng/mg Cr immediately after surgery and decreased to 8.04 [6.4–10.3] ng/mg Cr 24 hours after surgery (p<0.001). Post-hoc analysis confirmed that levels 24 hours after surgery were higher than those pre-operatively (p = 0.026). Infant patients had slightly higher median 8-iso-PGF2α immediately post-operatively (Fig 2A).

In regards to the type of surgery, there were no differences in pre-surgical levels of 8-iso-PGF2α between patients with CPB or without CPB (p = 0.4853), though CPB patients had higher 8-iso-PGF2α levels than no CPB patients immediately post-operation (p = 0.0214). Twenty-four hours after surgery, these differences were no longer observed (p = 0.9710) (Fig 2A). A regression analysis was performed to correct for age as a possible confounder, which showed no differences in the levels of 8-iso-PGF2α (p = 0.371).

Renal function was also analyzed as a possible confounder. There was no correlation between presurgical levels of creatinine and presurgical levels of 8-iso-PGF2 (p = 0.3925).

**Table 3. Differences in cSO2 values between CPB and no CPB patients.**

| | CPB | No CPB | P value |
|---|---|---|---|
| Intraoperative minutes spent <40% cSO2 | 21.5 [13–29] | 5.9 [3–9] | 0.0029 |
| Intraoperative minutes spent <50% cSO2 | 54.6 [40–68] | 19.7 [10–29] | 0.0004 |
| Intraoperative minutes spent outside of the 50–70% cSO2 range | 113.9 [96–131] | 74.77 [62–87] | 0.0014 |
| Intraoperative minutes spent >85% cSO2 | 16.2 [8–23] | 24.4 [14–35] | 0.1875 |
| 24 h post-operative minutes spent <40% cSO2 | 53 [13–94] | 106 [12–199] | 0.2402 |
| 24 h post-operative minutes spent <50% cSO2 | 183 [105–261] | 270 [123–416] | 0.2509 |
| 24 h post-operative minutes spent outside of the 50–70% cSO2 range | 659 [548–769] | 799 [643–956] | 0.1332 |
| 24 h post-operative minutes spent >85% cSO2 | 78 [24–133] | 86 [11–161] | 0.8656 |

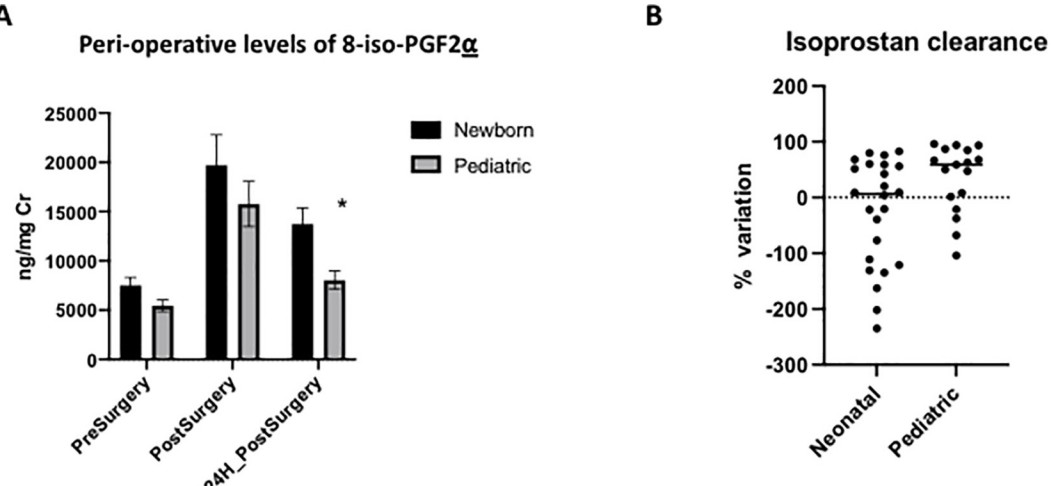

**Fig 2.** A: Peri-operative levels of 8-iso-PGF2α across all three time points by subgroup (Expressed in ng/mg Cr and Mean +/s SD). **B:** Isoprostanes Clearance [(Post-surgery 8-iso-PGF2α– 24 hours post-surgery 8-iso-PGF2α) ∗100]/ Post-surgery 8-iso-PGF2α. (∗ p <0.05).

Similarly, no correlation was between post-surgical levels of creatinine and immediate post-surgical levels of 8-iso-PGF2α (p = 0.8341), nor with levels of 8-iso-PGF2α at 24 hours post-surgery (p = 0.2251).

Amongst all CPB patients, no differences were found between newborns and infant patients in pre-surgical or immediately post-surgical levels (p = 0.1403 and 0.2204, respectively). There was, however, a significant difference in levels by age at 24 hours post-surgery (p = 0.0421) (Fig 3B).

Pre-surgical levels of isoprostanes were not influenced by patient characteristics (gender, gestational age, birth weight), clinical conditions (arterial umbilical pH, days of life at surgery, degree of cyanosis, aortic obstruction) or pre-surgical treatments (sub atmospheric treatment, prostaglandin infusion) (p>0.05).

Pre-surgical levels of OS among the three cyanosis groups (non-cyanotic, typical cyanotic and TGA) were not statistically different (p = 0.359), including in post-hoc analysis. In the post-operative period, however, there were significant differences between groups (p = 0.017);

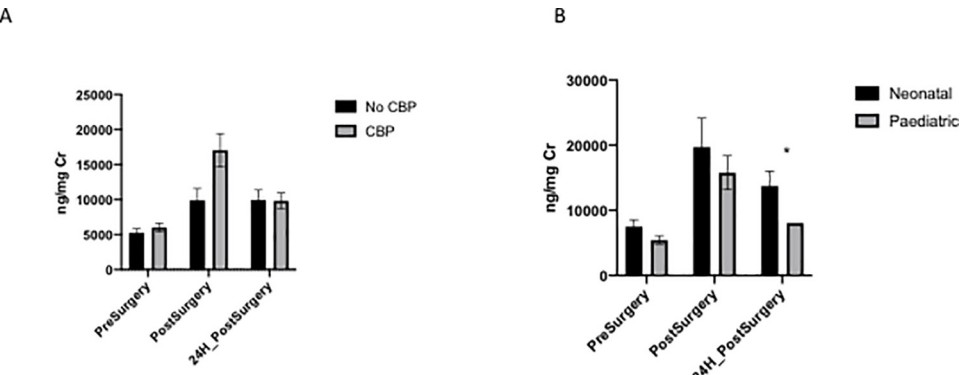

**Fig 3.** A: Isoprostane levels according to type of surgery. **B**: Isoprostanes levels only in CPB patients by age (Expressed in ng/mg Cr and Mean +/s SD). (∗ p <0.05).

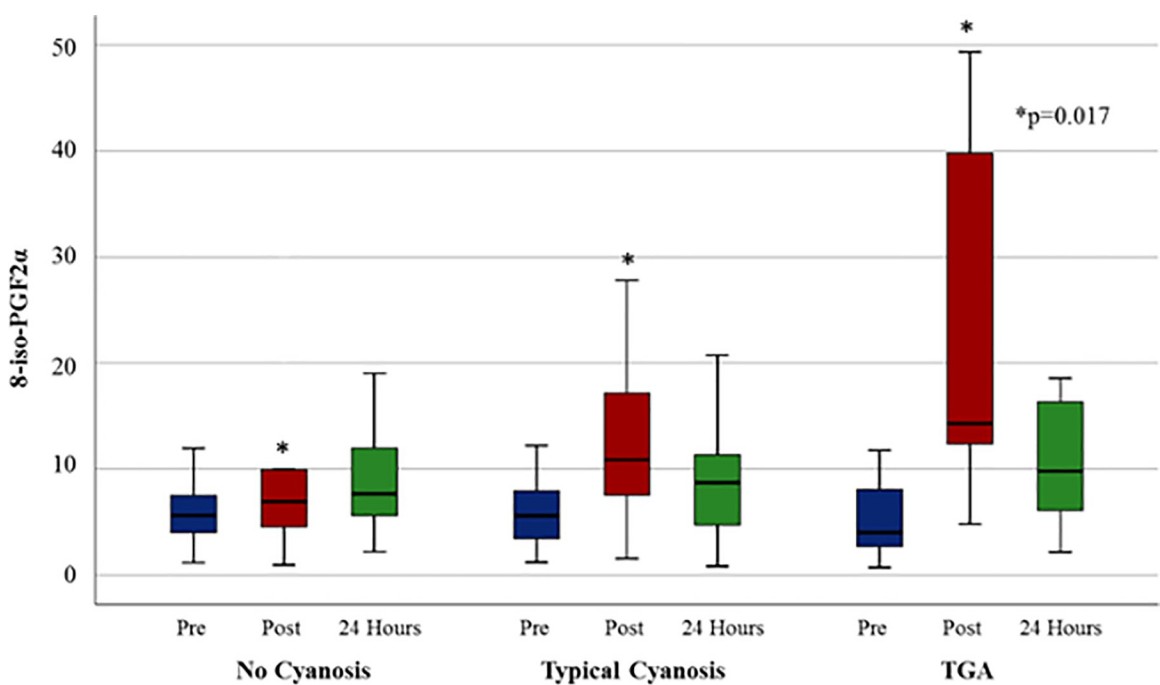

**Fig 4. Peri-operative levels of 8-iso-PGF2α by cyanotic state.** TGA patients had higher post-operative 8-iso-PGF2α than non-cyanotic and typically-cyanotic patients. (Expressed in ng/mg Cr and Mean +/s SD). (* p <0.05).

in post-hoc analysis of post-surgery OS levels, TGA patients' OS levels were higher compared to non-cyanotic patients (p = 0.019) and trended higher than typically-cyanotic patients (p = 0.095). No differences were found in OS levels at 24 hours (p = 0.9218) (Fig 4).

A regression analysis to exclude CPB effects in TGA patients as a confounder was performed, and the differences between groups persisted (p = 0.0214).

In neonatal patients, 8-iso-PGF2α immediately following surgery was inversely correlated with lower intra-operative temperature (rho = -0.572, p = 0.003). There was no correlation between OS and time of CPB, aortic cross-clamp time, lower hematocrit or quantity of ultrafiltrate in any subgroup.

Patients with intra-operative seizures had higher post-operative 8-iso-PGF2α levels than those without seizures (p = 0.027). There were no correlations between 8-iso-PGF2α levels and intra-operative background electroencephalographic patterns.

There were no correlations between intra-operative cSO2 values and post-operative 8-iso-PGF2α, though neonatal patients who experienced longer times of cSO2 below 50% within 24 hours post-operatively had significantly higher post-surgery 8-iso-PGF2α levels (p = 0.05).

8-iso-PGF2α clearance at 24 hours post-surgery was different between newborns and infant patients (p = 0.0489) (Fig 2B), though the difference did not reach statistical significance after correcting for type of surgery (p = 0.0768). All infant patients in our population required CPB surgery. When clearance of OS was analyzed only in neonatal patients, no differences were found in 8-iso-PGF2α levels between those with CPB and those without (p = 0.3392).8-iso-PGF2α clearance was inversely correlated with days of mechanical ventilation (p = 0.05), ICU LOS (p = 0.05) and VIS score at 24 hours (p = 0.036). In terms of NBO evaluations, no differences were found between global scores and the rate of 8-iso-PGF2α descent (p = 0.3243), however, sub-domain analysis revealed that the rate of 8-iso-PGF2α descent was inversely

correlated with habituation score (rho = -0.6688; p = 0.0244). All these results were corrected for age.

## Discussion

This study demonstrates an increase in post-operative oxidative stress response measured by post-operative urine levels of 8-iso-PGF2α in children undergoing heart surgery, and especially in those undergoing cardiopulmonary bypass. Moreover, our study reveals that patients with pre-operative cyanosis and specifically those with Transposition of the Great Arteries have the highest post-operative 8-iso-PGF2α levels after surgery.

We have also demonstrated that elevated isoprostane clearance, measured by the rate of 8-iso-PGF2α descent within the first 24 hours post-surgery, is an important clinical prognostic factor. A slower isoprostane clearance was inversely correlated with several post-operative clinical outcomes such as prolonged days of mechanical ventilation and ICU length of stay, higher VIS score at 24 hours, and poorer habituation capacity in NBO neurological evaluation. Infant patients had more rapid 8-iso-PGF2α clearance when compared to newborn patients, in accordance with the physiological immaturity of antioxidant systems during the neonatal period.

OS is characterized by an imbalance between increased free radicals and antioxidant defenses. Recently, OS has been recognized as being associated with many disease processes. Several *in vitro* markers of OS are available, but most have limited *in vivo* value because of low sensitivity and/or specificity. Isoprostanes have been proposed as one of the most reliable biomarkers to evaluate OS status *in vivo*, serving as an important tool to understand the role of OS in many conditions [13].

Pediatric heart surgery is associated with increased inflammation and the production of reactive oxygen species [1]. In our study cohort, mean levels of 8-iso-PGF2α pre- and post-surgery were elevated compared to reference values, although normal values have not been well established in pediatric patients [14]. The pathophysiology of OS during cardiac surgery involving CPB is multi-factorial. Systemic inflammation, mechanical shear stress, intra-operative changes in blood pressure and perfusion and significant hemodilution have been reported as possible triggers [1, 2, 15]. These effects can be even more profound in smaller patients given the higher proportion of the patient's blood volume exposed to the circuit. A previous study by Gil-Gomez et al. found a direct correlation between the time of extracorporeal circulation and malondialdehyde levels, another biomarker of lipid peroxidation [16]. In our cohort of patients, the most important variable influencing the OS response was the utilization of CPB. While OS has been shown to be more severe in CPB surgeries compared to non-CPB surgeries in adults, this is the first study to our knowledge to demonstrate that this pattern holds true in pediatric patients, as well [3, 17].

Interestingly, we did not find a correlation between the time of extracorporeal circulation and levels of 8-iso-PGF2α.

TGA patients undergoing arterial switch procedures experienced higher levels of post-operative 8-iso-PGF2α than patients with typical cyanosis and no cyanosis. Following periods of hypoxia, oxidative toxicity occurs during the reoxygenation phase and is thought to be mediated by the lipid peroxidation pathway [9]. Caputo et al.'s findings also supported this model, showing that controlled reoxygenation following CPB in pediatric patients led to lower levels of post-operative 8-iso-PGF2α compared to hyperoxic reoxygenation [18]. Therefore, the drastic changes in hemoglobin saturation and partial pressure of oxygen from pre-surgery to post-surgery that we identified in patients with TGA is consistent with their higher levels of post-operative OS. Despite the fact that the TGA patients in our cohort were younger at the time of surgery than other patients, their baseline 8-iso-PGF2α values were similar to those of

the typical cyanosis or no cyanosis groups. Furthermore, immediately post-operative OS levels of patients with TGA were statistically different when compared to non-cyanotic patients, and trended higher than those of typical-cyanotic patients. All these data suggest that during surgery, changes in oxygenation, perfusion, hemodilution, and other variables related to oxidative status had a greater effect in patients with a pre-surgical cyanotic condition, particularly those with TGA. Our results contrast those of Altin et al. who did not find any differences in OS between cyanotic and acyanotic patients following CPB surgery [19]. This discrepancy could be partially explained by patient population, as only 3 of the 30 patients in their cyanotic group had TGA anatomy.Patients who experienced intra-operative electrical seizures had significantly higher post-operative 8-iso-PGF2α levels than patients with no intra-operative seizure activity. These findings are in accordance with previous studies in rat models showing considerable lipid peroxidation as early as 10 minutes after seizure onset [20].

We discovered that neonatal patients who spent more time with cSpO2 below 50% within the first 24 hours following surgery experienced higher elevations of OS biomarkers compared to those with normal cerebral oxygenation. This could be an indication that reoxygenation occurring after extended periods of cerebral hypoxia contributed to delayed recovery from oxidative injury. Interestingly, we did not find any relationship between intra-operative time with cSO2 >85% and post-operative OS, unlike previous findings in adult cardiac surgery patients [21], though time spent in hyperoxic states was generally brief.

Previous studies have drawn mixed conclusions concerning the association of peri-operative OS and clinical outcomes [22, 23], however, our data revealed several significant correlations. The rate of normalization of 8-iso-PGF2α within the first 24 hours following surgery was inversely correlated with days of mechanical ventilation, ICU length of stay, inotropic support score and habituation score in NBO neurological evaluation. These findings may be related to a variety of factors. Albers et al. demonstrated a correlation between peri-operative 8-iso-PGF2α levels and markers of impaired ventilation in pediatric univentricular patients, which may have played a role in increased LOS [24]. Additionally, OS- induced endothelial cell dysfunction may have contributed to the requirement for increased inotropic support following surgery. Finally, there is an abundance of literature relating congenital heart disease surgery and neurodevelopmental disabilities. A prospective study published by Massaro et al reported suboptimal neurobehavioral performance in CHD patients after surgery [25], though no previously published papers have related OS and neurobehavior disabilities in this population.

Our study had several limitations. First of all, we only examined one biomarker of OS representing lipid peroxidation. Though 8-iso-PGF2α is generally accepted as one of the most reliable in vivo biomarkers of OS, several other OS pathways, as well as antioxidant states, could also play a role in the outcome of this population. In addition, our clinical evaluation was limited to the early post-operative period. Long term follow-up with neurocognitive evaluations and functional status will provide a more accurate assessment.

## Conclusions

In conclusion, children undergoing cardiac surgery, particularly neonatal CPB patients, experience significant post-operative OS as quantified by urine 8-iso-PGF2α levels, which might play an important role in postoperative morbidity. Among all, neonates with TGA undergoing arterial switch operations demonstrate the highest post-operative OS response. The impact of maturation on the oxidative stress response is important, especially in the youngest patients. In our population, neonatal patients (those less than one month old) show significantly slower recovery from oxidative stress. This may reflect the immaturity of the antioxidant system

during the first days of life, as has been reported in other pathologies. Moreover, our results suggest that rapid clearance of isoprostanes, which we found in older patients with more mature antioxidant systems, might be associated with better clinical outcomes.

Future studies encompassing larger series of patients should assess the prognostic value of the OS response in predicting long-term clinical and neurodevelopmental outcomes.

## Supporting information

**S1 Data.**
(DOCX)

**S1 File.**
(XLSX)

## Acknowledgments

The authors would like to acknowledge support from Anna Valls Lafon, Dr. Laura Carrara, the pediatric and neonatal ICU nurses at Hospital Sant Joan de Déu, and the *Vanderbilt Medical Scholars Program* (SH).

## Author Contributions

**Conceptualization:** Marta Camprubí Camprubí, Joan Sanchez-de-Toledo.

**Data curation:** Stephanie Hadley, Debora Cañizo Vazquez, Miriam Lopez Abad, Stefano Congiu, Dmytro Lushchencov.

**Formal analysis:** Marta Camprubí Camprubí.

**Funding acquisition:** Marta Camprubí Camprubí, Joan Sanchez-de-Toledo.

**Investigation:** Stephanie Hadley, Debora Cañizo Vazquez, Stefano Congiu, Dmytro Lushchencov.

**Methodology:** Miriam Lopez Abad.

**Supervision:** Marta Camprubí Camprubí, Joan Sanchez-de-Toledo.

**Writing – original draft:** Stephanie Hadley, Debora Cañizo Vazquez.

**Writing – review & editing:** Marta Camprubí Camprubí, Joan Sanchez-de-Toledo.

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
