## [Decision Letter · Decision Letter 0]

16 Dec 2020

PONE-D-20-35518

Oxidative Stress Responce in children Undergoing Cardiac Surgery: utility of the Isoprostans Clearence

PLOS ONE

Dear Dr. Camprubí Camprubí,

Thank you for submitting your manuscript to PLOS ONE. After careful consideration, we feel that it has merit but does not fully meet PLOS ONE’s publication criteria as it currently stands. Therefore, we invite you to submit a revised version of the manuscript that addresses the points raised during the review process.

We look forward to receiving your revised manuscript.

Kind regards,

Juan Carlos Lopez-Delgado, MD, PhD

Academic Editor

PLOS ONE

Additional Editor Comments:

Dear author,

Please, follow instructions from the two reviewers and use some sort of PRS before resubmission. English and format should be perfect before consider your manuscript for publication.

PLOS One does not have limits for tables and related information about the topic of your manuscript. Please, enlarge the information retrieved as much as possible. This is a small cohort and information would be used for future meta-analyses.

Both of the reviewers recommended reject. However, I would like to give you the chance to improve your manuscript in order to change the mind of the present reviewers.

Regards,

Juan Carlos

Journal Requirements:

2. In your Methods section, please ensure you have also stated whether you obtained consent from parents or guardians of the minors included in the study or whether the research ethics committee or IRB specifically waived the need for their consent.

Reviewers' comments:

Reviewer's Responses to Questions

**Comments to the Author**

1. Is the manuscript technically sound, and do the data support the conclusions?

Reviewer #1: Partly

Reviewer #2: Partly

2. Has the statistical analysis been performed appropriately and rigorously? 

Reviewer #1: Yes

Reviewer #2: Yes

3. Have the authors made all data underlying the findings in their manuscript fully available?

Reviewer #1: Yes

Reviewer #2: Yes

4. Is the manuscript presented in an intelligible fashion and written in standard English?

Reviewer #1: Yes

Reviewer #2: Yes

5. Review Comments to the Author

Reviewer #1: This is an interesting study that reports that infants undergoing cardiac surgery have oxidative stress as measured by increases levels of 8-iso-PGF2a in their urine before, during, an in the few days after surgery. When levels of this marker were compared to various clinical parameters and outcomes, they found that higher oxidative stress was correlated with use of cardiac bypass (CBP) and was higher in patients with d-TGA, who they say had extreme cyanosis before surgery. High levels of this marker are associated with longer ventilation and ICU days, increased inotropic support, and habituation score. They conclude that infants undergoing cardiac surgery have increased oxidative stress that may play a role in post-operative morbidity. However, there are many pieces of data missing in this report that make this report incomplete and difficult to interpret. This lessens enthusiasm for its publication in the current form.

Major issues:

1. The authors need to make a better argument that infants have poor “cardiovascular bypass tolerance.” It is unclear what they mean by this and the references they give seem only to discuss this in relation to oxidative stress, which, as they state, has uncertain relation to clinical outcomes. In fact, some data suggests that neonates may have better tolerance to cardiac bypass.

2. There are two groups presented, neonates and pediatric patients, and some neonates but all pediatric patients underwent CBP. Some analyses compare data based on age (neonate vrs pediatric) while other analyses are for CBP vrs no CBP. The analyses should consistently include both comparisons. For example, the authors state that the immature state of anti-oxidant defenses is revealed in their data and this is detrimental to infants undergoing surgery, but they state in the text (page 12, lines 215-218) say that this is not the case, while the data in Figure 2 suggests that CBP is a larger contributor to elevated oxidative stress. However, since all of the pediatric patients while only some neonates underwent CBP, it might be that the interaction of age and CBP would be revealing. In addition, might the data on d-TGA patients be significant because they underwent CBP?

3. More information should be given about the patient characteristics and state before surgery, such as comorbid conditions, clinical condition, other procedures, and the defects themselves with their surgical approach and need for bypass or not. Only limited data on CHD lesions is presented in Tables 1 and 2, but Table 2 should break down diagnoses by age category and which diagnoses did not receive cardiac bypass as the low amount of bypass in neonates seems odd. For example, they make much of the fact that patients with d-TGA have profound cyanosis that makes them more likely to have oxidative stress. However, such cyanosis is usually corrected with an atrial septostomy and treatment with prostaglandins, increasing systemic saturation and stabilizing the patient. Such patients are not usually rushed to the operating room when unstable. Furthermore, other congenital heart defects can present with cyanosis as profound as seen in d-TGA. Therefore, their explanation of why d-TGA patients had higher levels of 8-iso-PGF2a in the discussion (page 15) is speculative and not probably logical.

4. Levels of 8-iso-PGF2a are reported as ng/mg creatinine. However, at least creatinine if not other measures of renal function should be reported. Based on the data reported, one questions whether the clearance of 8-iso-PGF2a is dependent on renal function, and that all results reported in this manuscript could be due to low cardiac output yielding poor renal function and thus slow clearance of 8-iso-PGF2a. However, I do recognize that such as state might also prolong systemic oxidative stress for the reasons given by the authors.

5. Although cSO2 is reported, arterial saturation and p02 and measures of cardiac output are not reported. Not only might cSO2 depend on these parameters, but these values might also reflect systemic oxygen delivery, which could affect oxidative stress throughout the body. This would also clear up the descriptions of “typical cyanosis” and “extreme cyanosis” that are used in the text.

6. Very little is presented on the EEG data, and no correlation is made to patient characteristics, including age and use of CBP, which may be important relations.

7. The figure legends need to be expanded to actually explain the Figures, including mentioning statistical and graphical parameters (N, +/- SD or SEM, statistical test used). Furthermore, only Figure 3 has markers of statistical significance (but with the N of each group not presented), and it is reported in the text that the post-surgery 8-iso-PGF2a levels are significantly different in Figure 1a, but given the data presented it is hard to believe this without more information.

Minor issues:

1. There are some spelling and grammatical errors. The manuscript should be gone over to eliminate these. Some examples include:

a. “response,” “isoprostanes,” and “clearance” are mis-spelled in the title.

b. Page 2, line 40, “measure” should be “measured”

c. Page 3, line 63, “increase” should be “increased” or “increase in”

d. Page 4, Line 91, “Material ad methods” should be “Materials and methods”

e. Page 6, line 120, “Continues” should be “Continuous”

f. Page 10, line 199, “were exposed” should be something like “are evident”

g. There are other examples

2. The statistics section of the materials and methods states that data were checked for normality, but the only multiple comparison test mentioned was Kruskal-Wallace. Was all data non-parametric?

3. In the text, page 10, lines 197-198, the lower cSO2 values are reported to be 68% while the higher are 60%; please correct this error.

4. The authors should explain what they mean by the statement “These results are not surprising given the smaller size of the patients in relation to the bypass circuit. . .” (Page 15, lines 286-287).

Reviewer #2: This manuscript examines the role of oxidative stress (OS) in the post-op course of neonates and infants with congenital cardiac defects. The topic is an important one as oxidative stress may be a critical determinant of post-operative outcomes and may be a therapeutic target in the strategy to improve outcomes. The choice of 8-isoprostaglandine-F2α as a readout of oxidative stress exposure is reasonable. The cohort is relatively small in size and diverse, complicating interpretation of the findings. Comparison of the two subgroups, the neonates and the infants, is heavily confounded by different clinical factors and operative approaches. For instance, only a third of the neonates underwent CPB which will have an important effect on inflammatory activation, endothelial injury and oxidative stress. The description of profound cyanosis and acidosis in the dTGA patients is concerning regarding how those patients were managed. Commonly, balloon atrial septostomy is performed to prevent severe cyanosis and acidosis and a period of stabilization is allowed before surgery. In addition, the surgical complexity is relatively low with most patients being STAT category 1 for surgical complexity which will affect oxidative stress burden.

Major concerns:

1. The neonatal and pediatric patient populations are different enough that any statements regarding reflection effect of age on oxidative stress management and capacity in this study must be made with caution.

2. “Typical” and Extreme” cyanosis is not well defined and the management of dTGA patients with regard to frequency of and indications for BAS are not described.

3. The number of patients requiring deep hypothermic circulatory arrest is not described.

4. Particularly for the neonatal cohort, it would be important to know if there were any that were born prematurely as this may affect both OS response and clinical outcomes.

5. To more directly compare the neonatal and pediatric cohorts, a comparison of OS levels and clearance should be performed specifically in those neonatal subjects who underwent CPB.

6. In the subdomain analyses, 8-isoprostaglandine-F2α descent is likely confounded with CPB exposure in the neonates and it is unclear from the data presented if the effects are due to OS or other effects of CPB.

Minor comments:

1. In abstract, the “(8.04 ng/mg Cr were higher than those pre-operatives 5.7 [6.4-10.3]…” is missing a ).

2. Typo in Methods and Materials

3. The clearance formula is not properly displayed.

4. In line 197, the “lower cSO2 values…” line does not make sense as written.

6. PLOS authors have the option to publish the peer review history of their article (what does this mean?). If published, this will include your full peer review and any attached files.

Reviewer #1: No

Reviewer #2: No

---

## [Author Response · Author response to Decision Letter 0]

3 Feb 2021

Reviewer #1: 

Major issues:

1. The authors need to make a better argument that infants have poor “cardiovascular bypass tolerance.” It is unclear what they mean by this and the references they give seem only to discuss this in relation to oxidative stress, which, as they state, has uncertain relation to clinical outcomes. In fact, some data suggests that neonates may have better tolerance to cardiac bypass.

The reviewer is right, pediatric patients (including neonates) may have a good tolerance to cardiac bypass. What we were trying to state was that Cardiopulmonary Bypass (CPB) could increase the oxidative stress response, as we have proved in our paper. The sentence that was written in the paper could lead to a misunderstanding, so we have rephrased to make it clearer. (lines 117-119)

2. There are two groups presented, neonates and pediatric patients, and some neonates but all pediatric patients underwent CPB. Some analyses compare data based on age (neonate vrs pediatric) while other analyses are for CPB vrs no CPB. The analyses should consistently include both comparisons. For example, the authors state that the immature state of anti-oxidant defenses is revealed in their data and this is detrimental to infants undergoing surgery, but they state in the text (page 12, lines 215-218) say that this is not the case, while the data in Figure 2 suggests that CPB is a larger contributor to elevated oxidative stress. However, since all of the pediatric patients while only some neonates underwent CPB, it might be that the interaction of age and CPB would be revealing. In addition, might the data on d-TGA patients be significant because they underwent CPB?

Thank you for your comment. As we have explained with some of the included references ( Torres-Cuevas et al, 2017), the antioxidant system is very immature during the first month of life. Consequently, the antioxidant response is decreased, and the oxidative damage seems to be more important during this period, achieving its maturity progressively. 

Moreover, CPB surgery seems to be one of the most important factors contributing to oxidative stress damage. Despite the fact that patient age seems to be a protective factor, the older the patient at time of surgery, the lower the oxidative stress response. To improve the understanding of these results, and following the suggestions of the reviewers, we have included a better explanation of this and a complementary figure (Figure 2b) to make it more understandable. Included in line 340 to 351.

“A regression analysis was performed to correct for age as a possible confounder, which showed no differences in the levels of 8-iso-PGF2α (p=0.371). Amongst all CPB patients, neonates had overall increased OS response both pre- and post-surgery compare to pediatric patients with differences reaching statistical significance at 24 hours post-surgery (p=0.0421) (Fig 2B)”. 

Another interesting point that the reviewer suggested is the effect of CPB on d-TGA patients since all of them required CPB surgery for correction. A multivariable regression to exclude CPB effects as a confounder has been performed, and the differences between groups persists (p=0.0214). To clarify that, we have also included the analysis of OS considering cyanosis classification only in patients that require CPB. 

This new analysis has been included in lines 366-367. 

“A regression analysis to exclude CPB effects in TGA patients as a confounder was performed, and the differences between groups persisted (p=0.0214). “

3. More information should be given about the patient characteristics and state before surgery, such as comorbid conditions, clinical condition, other procedures, and the defects themselves with their surgical approach and need for bypass or not. Only limited data on CHD lesions is presented in Tables 1 and 2, but Table 2 should break down diagnoses by age category and which diagnoses did not receive cardiac bypass as the low amount of bypass in neonates seems odd. For example, they make much of the fact that patients with d-TGA have profound cyanosis that makes them more likely to have oxidative stress. However, such cyanosis is usually corrected with an atrial septostomy and treatment with prostaglandins, increasing systemic saturation and stabilizing the patient. Such patients are not usually rushed to the operating room when unstable. Furthermore, other congenital heart defects can present with cyanosis as profound as seen in d-TGA. Therefore, their explanation of why d-TGA patients had higher levels of 8-iso-PGF2a in the discussion (page 15) is speculative and not probably logical.

Considering reviewer suggestions, some pre-surgical patient characteristics have been included in table 1. Moreover, table 2 has been rebuilt showing diagnosis by age category and kind of surgery.

“Pre-surgical levels of OS among the three cyanosis groups (non-cyanotic, mild-cyanotic and TGA) were not statistically different (p=0.359), including in post-hoc analysis, confirming that the basal levels of OS biomarkers were similar among these groups before surgery. The differences appeared in the post-operative period. Those patients with a severe degree of cyanosis caused by Transposition of the Great Arteries had the highest post-operative 8-iso-PGF2α levels (p=0.017). Moreover, in the post-hoc analysis of post-surgery OS levels, TGA patients’ OS levels were higher when compared to non-cyanotic patients (p=0.019) and there was a tendency when compared to mildly-cyanotic patients (p=0.095). No differences were found in OS levels at 24 hours (p=0.9218) (Fig 3)”.

“All these data suggest that during surgery, those changes in oxygenation, perfusion, hemodilution, and other variables related to oxidative status had a greater effect in those patients with a pre-surgical cyanotic condition, especially in those with TGA.”

To clarify this, we have included a more accurate statistical description of the results in lines 356-366, and we have also included additional explanations in the discussion lines 572-575.

4. Levels of 8-iso-PGF2a are reported as ng/mg creatinine. However, at least creatinine if not other measures of renal function should be reported. Based on the data reported, one questions whether the clearance of 8-iso-PGF2a is dependent on renal function, and that all results reported in this manuscript could be due to low cardiac output yielding poor renal function and thus slow clearance of 8-iso-PGF2a. However, I do recognize that such as state might also prolong systemic oxidative stress for the reasons given by the authors.

We report levels of 8-iso-PGF2a in “ng/mg creatinine” following the instructions of the ELISA kit. All levels of proteins in urine must be normalized by another protein, to prevent dilutional influences. 8-iso-PGF2 levels were usually normalized with urinary levels of creatinine. With this normalization the effect of ultrafiltration or possible renal function alterations were avoided. 

5. Although cSO2 is reported, arterial saturation and p02 and measures of cardiac output are not reported. Not only might cSO2 depend on these parameters, but these values might also reflect systemic oxygen delivery, which could affect oxidative stress throughout the body. This would also clear up the descriptions of “typical cyanosis” and “extreme cyanosis” that are used in the text.

We completely agree with the reviewer’s comments. Unfortunately, data on preoperative oxygen saturation and continues hemodynamics are not available since the study data was collected before the upgrade of the unit monitoring system as well as the implementation of the new electronic medical report platform. Both systems would have helped to better understand the underlying CHD and the preoperative oxygenation state.

Cyanosis in congenital heart disease is due to the existence of either intra- or extra-cardiac shunts resulting in deoxygenated blood entering the systemic circulation and thus, generating different levels of cyanosis. Cyanotic congenital heart disease (CCHD) has been previously classified as follows: 1) CCHD with decreased blood flow, typically classified as tetralogy of Fallot physiology (typical cyanosis) and; 2) Cyanotic congenital heart disease with increased blood flow, typically classified as transposition physiology (extreme cyanosis). 

We have included in the updated version of the manuscript an explanation of this classification published by Rohit & Shrivastava in 2018, to clarify this topic (lines 146-151)

6. Very little is presented on the EEG data, and no correlation is made to patient characteristics, including age and use of CPB, which may be important relations.

According to the reviewer suggestion some more data about aEEG have been included in the text. aEEG patterns and the presence of electrical seizures has been analyzed in relation to CPB and age. This explanation appears in lines 273-278.

7. The figure legends need to be expanded to actually explain the Figures, including mentioning statistical and graphical parameters (N, +/- SD or SEM, statistical test used). Furthermore, only Figure 3 has markers of statistical significance (but with the N of each group not presented), and it is reported in the text that the post-surgery 8-iso-PGF2a levels are significantly different in Figure 1a, but given the data presented it is hard to believe this without more information.

Figure and figure legends have been modified. Statistical significance has been included and some more explanations too.

Minor issues:

1. There are some spelling and grammatical errors. The manuscript should be gone over to eliminate these. Some examples include:

a. “response,” “isoprostanes,” and “clearance” are mis-spelled in the title.

b. Page 2, line 40, “measure” should be “measured”

c. Page 3, line 63, “increase” should be “increased” or “increase in”

d. Page 4, Line 91, “Material ad methods” should be “Materials and methods”

e. Page 6, line 120, “Continues” should be “Continuous”

f. Page 10, line 199, “were exposed” should be something like “are evident”

g. There are other examples

Thanks for the corrections, all of them have been changed in the text and we have also reviewed the manuscript carefully again to correct other errors.

2. The statistics section of the materials and methods states that data were checked for normality, but the only multiple comparison test mentioned was Kruskal-Wallace. Was all data non-parametric?

Yes, most of the analyzed variables were non-parametric. Despite that, previous to the analysis their normality was checked, and we included it in the statistics description. 

3. In the text, page 10, lines 197-198, the lower cSO2 values are reported to be 68% while the higher are 60%; please correct this error.

Thanks for the suggestion, we have changed it in the text.

4. The authors should explain what they mean by the statement “These results are not surprising given the smaller size of the patients in relation to the bypass circuit. . .” (Page 15, lines 286-287).

This statement is a little bit confusing, so following the reviewer’s suggestion we have changed it. 

Reviewer #2: 

Major concerns:

1. The neonatal and pediatric patient populations are different enough that any statements regarding reflection effect of age on oxidative stress management and capacity in this study must be made with caution.

The reviewer is right, and this is one of the hot topics of our paper. The effect of maturation over the oxidative stress response is of paramount importance, especially in the youngest patients. In our population, neonatal patients (those less than one month old) show significant differences in the clearance of Isoprostanes. This may reflect the immaturity of this system during the first days of life, as it also has been reported in other pathologies

Another point that is also interesting regarding age is that no statistical differences were detected in OS levels prior to surgery or immediately after surgery. Considering these data, we could hypothesize that the basal and post-surgery responses are similar in both age groups, and that the major differences appear when the antioxidant systems have to be used, as these systems are more efficient in the older patients.

A final conclusion about this point has been included in the paper line 647-653.

2. “Typical” and Extreme” cyanosis is not well defined and the management of dTGA patients with regard to frequency of and indications for BAS are not described.

Considering reviewer suggestion we have included a better explanation of the classification .

Cyanosis in congenital heart disease is due to the existence of either intra- or extra-cardiac shunts resulting in deoxygenated blood entering the systemic circulation and thus, generating different levels of cyanosis. Cyanotic congenital heart disease (CCHD) has been previously classified as follows: 1) CCHD with decreased blood flow, typically classified as tetralogy of Fallot physiology (typical cyanosis) and; 2) Cyanotic congenital heart disease with increased blood flow, typically classified as transposition physiology (extreme cyanosis). 

We have included in the updated version of the manuscript an explanation of this classification published by Rohit & Shrivastava in 2018, to clarify this topic (lines 146-151)

Frequency and indication of BAS have been added in Material and Methods as well as in table 1.

3. The number of patients requiring deep hypothermic circulatory arrest is not described.

We have included these data in Material and Methods line 200-201.

4. Particularly for the neonatal cohort, it would be important to know if there were any that were born prematurely as this may affect both OS response and clinical outcomes.

Thanks for the suggestion, these data have been included in table 1.

5. To more directly compare the neonatal and pediatric cohorts, a comparison of OS levels and clearance should be performed specifically in those neonatal subjects who underwent CPB.

Considering the reviewer suggestion, this analysis has been performed and included in the text lines 348-351 and in Figure 2B.

6. In the subdomain analyses, 8-isoprostaglandine-F2α descent is likely confounded with CPB exposure in the neonates and it is unclear from the data presented if the effects are due to OS or other effects of CPB.

All pediatric patients in our population require CPB surgery. When clearance of OS was analyzed only in neonatal patients to consider the effect of 8-isoprostaglandine-F2α descent, no differences were found (p=0.3392)

Thanks for the suggestion, we have clarified this point in the text, lines 500-503.

Minor comments:

1. In abstract, the “(8.04 ng/mg Cr were higher than those pre-operatives 5.7 [6.4-10.3] …” is missing a ).

Thanks, this has been changed in the text.

2. Typo in Methods and Materials

Thanks, this has been corrected in the updated version of the manuscript. 

3. The clearance formula is not properly displayed.

Thanks, this has been modified in the updated version of the manuscript. 

4. In line 197, the “lower cSO2 values…” line does not make sense as written.

Thank you for pointing at this. We have identified the grammatical mistake and we have rephrased the sentence in the updated version of the manuscript.

---

## [Decision Letter · Decision Letter 1]

17 Feb 2021

PONE-D-20-35518R1

Oxidative Stress Response in Children Undergoing Cardiac Surgery: Utility of the Clearence of Isoprostanes

PLOS ONE

Dear Dr. Camprubí Camprubí,

Thank you for submitting your manuscript to PLOS ONE. After careful consideration, we feel that it has merit but does not fully meet PLOS ONE’s publication criteria as it currently stands. Therefore, we invite you to submit a revised version of the manuscript that addresses the points raised during the review process.

We look forward to receiving your revised manuscript.

Kind regards,

Juan Carlos Lopez-Delgado, MD, PhD

Academic Editor

PLOS ONE

Additional Editor Comments (if provided):

Please, address the concerns described by Reviewer 1. All of them should be answered properly.

Secondly, send to a native english speaker and/or provide a review of the manuscript to avoid english errors (e.g., proof reading services). Provide also a proof of english correction.

Thank you very much for the great job you have done until now.

Reviewers' comments:

Reviewer's Responses to Questions

**Comments to the Author**

1. If the authors have adequately addressed your comments raised in a previous round of review and you feel that this manuscript is now acceptable for publication, you may indicate that here to bypass the “Comments to the Author” section, enter your conflict of interest statement in the “Confidential to Editor” section, and submit your "Accept" recommendation.

Reviewer #1: (No Response)

Reviewer #2: All comments have been addressed

2. Is the manuscript technically sound, and do the data support the conclusions?

Reviewer #1: Partly

Reviewer #2: Yes

3. Has the statistical analysis been performed appropriately and rigorously? 

Reviewer #1: Yes

Reviewer #2: Yes

4. Have the authors made all data underlying the findings in their manuscript fully available?

Reviewer #1: (No Response)

Reviewer #2: Yes

5. Is the manuscript presented in an intelligible fashion and written in standard English?

Reviewer #1: Yes

Reviewer #2: Yes

6. Review Comments to the Author

Reviewer #1: This remains an interesting study that reports that infants undergoing cardiac surgery have oxidative stress as measured by increases levels of 8-iso-PGF2a in their urine before, during, an in the few days after surgery. When levels of this marker were compared to various clinical parameters and outcomes, they found that higher oxidative stress was correlated with use of cardiopulmonary bypass (CPB) in neonates immediately after surgery and was higher in patients with d-TGA. High levels of this marker are associated with longer ventilation and ICU days, increased inotropic support, and habituation score. They conclude that infants undergoing cardiac surgery have increased oxidative stress that may play a role in post-operative morbidity and that this is age dependent—older infants (>1 month of age) are less likely to have elevated oxidative stress as evaluated by urine 8-iso-PGF2. Although the authors have been responsive to the two reviewers, there remain problems with this work that need to be addressed. Please also note that some of these points were commented on by both reviewers independently.

1. The authors should reconsider their use of pediatric, infant, and neonate to describe patient groups. Pediatric is a more general term, whereas the authors are really evaluating a difference between neonates < 1 month of age and older infants.

2. Explanations of some of the comparisons remains remain unclear. For example, the statistics reported in lines 242-248 and in Figure 3 are confusing. What is the test used to compare post-operative levels of TGA patients to get a p=0.017 and to what are they compared? It is then stated that in "post-hoc" analysis, TGA patients are different from non-cyanotic (p=0.019) and mildly cyanotic (p=0.095) patients. How is this different from the previous sentence?

3. Cyanosis: Note that both reviewers questioned the authors’ use of cyanosis levels, and I continue to find the distinction of levels of cyanosis not appropriate. In the first paragraph of the Discussion, which is a nice summary of the results, the clinical entity of TGA is said to be different from other diagnoses, but throughout the text, the authors continue to use the terms mild/typical and severe/extreme cyanosis. Although they provide a review reference for this (which I could not get), their use of the terms (d-TGA is extreme cyanosis while all other cyanotic lesions are not) is not standard and will be confusing to the general reader and discounted by most pediatric cardiologists. Many patients with d-TGA have less cyanosis that those with the other lesions listed in Table 2, and some of them (HLHS, TAPVR) can have cyanosis AND increased pulmonary blood flow, like in d-TGA. If the authors are going to define levels of cyanosis, then they should report % hemoglobin saturations or partial pressures of oxygen. The explanation that these data are not available due to the lack of an electronic medical record is nonsense. Pre- and intra-operative saturations and blood gasses should be present in the paper medical record, like other reported data (e.g., intra-operative temperature). The fact that those with d-TGA have higher levels of OS in the immediate post-operative period is interesting and should be reported, but is it also interesting that OS in patients with d-TGA is not significantly different from other neonates with “typical cyanosis,” making one wonder if perhaps there is a correlation between actual cyanosis (saturation, pO2) and post-op OS. Therefore, I believe you should get these data, as the level of pre-operative (and post-operative for that matter) cyanosis could play a role in OS, as the authors point out in the discussion about oxygenation during reperfusion after surgery.

4. Table 1 needs to be modified.

a. The cell for STAT category needs to be justified to the left.

b. There should be more explanation somewhere about the surgeries performed (perhaps as a supplement). Although it is now nice that table 2 describes the diagnoses, based on these diagnoses, I would expect that more patients would have undergone cardiac bypass, unless there is a higher percentage than expected based on current care of patients undergoing palliative placement of systemic to PA shunts. Therefore, more information than a category of “Other” should be provided.

c. Please explain why so many neonates were treated preoperatively with subatmospheric oxygen.

5. The definition of abnormal intra-operative EEG patterns needs to be stated, since there were no differences in seizures.

6. In lines 207-208, should the sentence start: "The median cSO2 AFTER surgery. . ."

7. It appears the the CBP and no CBP headings in Table 3 are misplaced, since the text states that the mean sats for CBP and no CBP were 68 versus 60% while the numbers in the table reflect the opposite condition.

8. The authors have disregarded my comment about renal function and clearance. This should be addressed. 8-iso-PGF2a levels will depend on production and renal clearance (presumably there are no other methods of clearance), so if renal function is decreased, then this could explain elevated levels. It is not unusual that infants suffered acute kidney injury after cardiac surgery, and the effects of this may not be present for a few days because monitoring markers other than BUN and creatinine of this are only being investigated now. Therefore, the authors should report if the patients suffered clinical AKI and the creatinine levels independent of 8-iso-PGF2a levels.

Minor issues:

1. There are still some spelling and grammatical errors.

Reviewer #2: There remain some inherit limitations based on the cohort but the authors have done a nice job of addressing the concerns raised in the previous review. It will be hard to know whether slower clearance really is slower clearance or ongoing production.

7. PLOS authors have the option to publish the peer review history of their article (what does this mean?). If published, this will include your full peer review and any attached files.

Reviewer #1: No

Reviewer #2: No

---

## [Author Response · Author response to Decision Letter 1]

29 Mar 2021

Answers have been included in Responce to Reviewers

---

## [Editor Report · Decision Letter 2]

31 Mar 2021

Oxidative Stress Response in Children Undergoing Cardiac Surgery: Utility of the Clearence of Isoprostanes

PONE-D-20-35518R2

Dear Dr. Camprubí Camprubí,

We’re pleased to inform you that your manuscript has been judged scientifically suitable for publication and will be formally accepted for publication once it meets all outstanding technical requirements.

Kind regards,

Juan Carlos Lopez-Delgado, MD, PhD

Academic Editor

PLOS ONE

Additional Editor Comments (optional):

N/A
---

## [Editor Report · Acceptance letter]

23 Jun 2021

PONE-D-20-35518R2 

Oxidative Stress Response in Children Undergoing Cardiac Surgery: Utility of the Clearance of Isoprostanes 

Dear Dr. Camprubí Camprubí:

I'm pleased to inform you that your manuscript has been deemed suitable for publication in PLOS ONE. Congratulations! Your manuscript is now with our production department. 

Kind regards, 

on behalf of

Dr. Juan Carlos Lopez-Delgado 

Academic Editor

PLOS ONE